# NatB Catalytic Subunit Depletion Disrupts DNA Replication Initiation Leading to Senescence in MEFs

**DOI:** 10.3390/ijms24108724

**Published:** 2023-05-13

**Authors:** Jasmin Elurbide, Beatriz Carte, Joana Guedes, Rafael Aldabe

**Affiliations:** 1Division of Gene Therapy and Regulation of Gene Expression, Centre for Applied Medical Research CIMA, University of Navarra, 31008 Pamplona, Spain; jelurbide@alumni.unav.es (J.E.); bcarte@unav.es (B.C.); joanacpg18@gmail.com (J.G.); 2Department of Biology, Centre of Molecular and Environmental Biology (CBMA/UM), University of Minho, 4710-057 Braga, Portugal

**Keywords:** N-terminal acetylation, NatB, cell cycle, DNA replication, senescence, N-degron pathway, N-recognins, actin cytoskeleton

## Abstract

Alpha-aminoterminal acetyltransferase B (NatB) is a critical enzyme responsible for acetylating the aminoterminal end of proteins, thereby modifying approximately 21% of the proteome. This post-translational modification impacts protein folding, structure, stability, and interactions between proteins which, in turn, play a crucial role in modulating several biological functions. NatB has been widely studied for its role in cytoskeleton function and cell cycle regulation in different organisms, from yeast to human tumor cells. In this study, we aimed to understand the biological importance of this modification by inactivating the catalytic subunit of the NatB enzymatic complex, *Naa20*, in non-transformed mammal cells. Our findings demonstrate that depletion of NAA20 results in decreased cell cycle progression and DNA replication initiation, ultimately leading to the senescence program. Furthermore, we have identified NatB substrates that play a role in cell cycle progression, and their stability is compromised when NatB is inactivated. These results underscore the significance of N-terminal acetylation by NatB in regulating cell cycle progression and DNA replication.

## 1. Introduction

Protein biosynthesis is an energetically expensive process that requires high fidelity to ensure proper functioning of the resulting peptides [1]. To achieve this, cells have developed various strategies, including the use of modifying enzymes that introduce modifications at the aminoterminal (N-terminal, Nt) end of nascent polypeptides. These modifications can affect folding properties, protein interactions, and half-life, ultimately increasing the diversity of expressed proteins [2]. One of the most common modifications is N-terminal acetylation (Nt-acetylation, Nt-Ac), which involves covalently binding an acetyl group to the free alpha-amino group at the N-terminal end of the polypeptide [3]. Nt-acetylation is catalyzed by enzymes known as N-terminal acetyltransferases (NATs), of which there are currently eight different types (NatA-H) in eukaryotic organisms. The substrate specificity of these enzymes is determined by the first two amino acids of the polypeptide chain [4]. The NatB complex, which is composed of the catalytic subunit NAA20 and the auxiliary subunit NAA25, specifically acetylates the initial methionine when it is followed by an aspartic acid, glutamic acid, asparagine, or glutamine. Interestingly, when any of the subunits of the NatB complex are depleted or inactivated, it affects actin cytoskeleton function and delays cellular growth in yeast, *Drosophila melanogaster*, and human tumor cells [5,6,7,8,9].

Nt-acetylation has the potential to enhance protein stability by promoting correct protein folding and interaction with other cellular components, leading to the formation of functional complexes [10,11,12,13,14,15]. However, it can also serve as a signal for degradation via the Ac/N-degron pathway [16,17,18]. The N-degron pathways are responsible for regulating protein half-life by identifying and destabilizing N-terminal residues as degradation signals, or N-degrons, and promoting their proteasomal degradation. Thus, the N-degron pathways function as a sensor of the protein’s functional state [19,20]. Specific E3 ubiquitin ligases, known as N-recognins that depend on the identity of the N-terminal end to recognize N-degrons, promoting their polyubiquitination and proteasomal degradation [20,21,22]. Different branches of the N-degron pathways have been identified, depending on the N-degron’s identity. The Ac/N-degron pathway, for example, targets acetylated N-terminal residues such as Met, Ala, Val, Ser, Thr, and Cys, as well as Pro or Gly to a lesser extent, for degradation. Two E3 ubiquitin ligases, the RING E3 ligase March6 (Teb4) and Not4, a component of the Ccr4–Not complex, are responsible for the degradation of these acetylated N-terminal residues [20]. The N-degron pathway is responsible for regulating numerous cellular processes, including cell cycle and cytoskeleton function, among others [20,23]. Interestingly, research has established a correlation between the Ac/N-degron pathway and the regulation of blood pressure achieved through controlling the turnover of Rgs2 protein. Certain patients with hypertension exhibit the substitution of the second amino acid in Rgs2 with residues that destabilize the protein. This mutation triggers degradation via both the Ac/N-degron pathway, like the wild-type version, and the Arg/N-degron pathway. Consequently, Rgs2 levels diminish significantly, leading to hyperactivation of the Gq protein, which ultimately causes vasoconstriction, thereby demonstrating the crucial role of Nt-acetylation in regulating protein half-life [16,20,24].

The cell cycle is a well-organized, unidirectional process that ensures proper cell division [25]. To promote genomic stability and prevent DNA damage from being passed on to daughter cells, several pathways and checkpoints monitor cell cycle transitions from DNA replication to mitosis. Protein stability plays a crucial role in checkpoint progression, as seen in the case of pRb and p53. The accumulation of CyclinD-CDK4/6 and CyclinE-CDK2 promotes Rb hyperphosphorylation and degradation, releasing E2F from Rb inhibition and inducing E2F-dependent expression of genes important for cell cycle progression and DNA replication [25]. DNA replication can face disruption by both endogenous sources, such as reactive oxygen species, and exogenous sources, such as UV light or chemical toxins. Such disruptions lead to replication stress, which can impede the progress of the replication fork and even cause DNA breaks [26,27,28]. The presence of DNA lesions activates the DNA damage response (DDR), which involves the activation of various kinases that phosphorylate different proteins, such as protein H2A.X at Ser139, also known as γ-H2A.X, which serves as a platform for the recruitment of repair proteins to the DNA lesion [27,29]. Additionally, the DDR promotes the stabilization and activation of p53, which induces the CDK inhibitor p21 among other substrates, reducing cyclin/CDK activity and arresting the cell cycle to repair the DNA. If the DDR persists, it can cause a permanent cell cycle arrest, leading to cellular senescence and preventing the cell from responding to mitogenic or growth factor stimuli [30].

Senescent cells are known to remain viable despite alterations in their metabolic activity and resistance to apoptosis. These cells exhibit a diverse phenotype and undergo various cellular and molecular changes depending on the stimuli that promote the stable cell cycle arrest. In senescent cells, genes involved in growth factor signaling and cell cycle progression are downregulated, while genes encoding factors that promote cell cycle arrest are upregulated. The complexity of this phenomenon highlights the need for further research to fully understand the mechanisms underlying cellular senescence [31,32]. In the process of cellular senescence, lysosomal activity increases and there is a temporal cascade in the development of the complex senescence-associated secretory phenotype (SASP). This phenotype secretes various chemokines, pro-inflammatory cytokines, growth factors, and matrix-remodeling enzymes that define the SASP, and is regulated at the transcriptional level. Senescent cells may also undergo morphological and structural changes, such as an enlarged, flattened, multinucleated morphology with enlarged vacuoles, altered composition of the plasma membrane, and remarkable nuclear enlargement associated with chromatin protrusions or nuclear blebbing into the cytoplasmic [31,32]. These changes play essential roles in normal development and limit tumor progression by preventing potentially dysfunctional, damaged, or transformed cells from passing on their genomes to the next generation [32].

Our research aimed to understand how non-transformed mammalian cells are affected by the depletion of Naa20. The results showed that this led to the activation of senescence and a decrease in DNA replication initiation. This decrease was accompanied by an increase in DNA lesions, which indicates the importance of N-terminal acetylation in regulating the stability of NatB substrates involved in cell cycle progression. Our findings suggest that the NatB enzymatic complex plays a crucial role in mammalian cells and has potential implications in various biological processes.

## 2. Results

### 2.1. NatB Enzyme Is Required for Proliferation and DNA Synthesis

The NatB complex is integrated by the catalytic subunit NAA20 and the auxiliary subunit NAA25, being necessary for the interaction between both subunits for proper enzymatic function. To explore the biological relevance of NatB in mammal cells, we inactivated *Naa20* in mouse embryonic fibroblasts (MEFs). To generate Naa20-defective MEFs (*Naa20^−/−^*), we infected MEFs with a recombinant adenovirus 5 (Ad5) expressing the Cre recombinase under the constitutive promoter CMV (AdCre). Alternatively, we infected MEFs with an Ad5 lacking the Cre recombinase to obtain control MEFs (Naa20 WT). Inactivation of Naa20 in MEFs was effective just 3 days after AdCre inoculation, and this effect was maintained until the end of the experiment 3 days later (Appendix A). Interestingly, at the end of the experiment there was almost a complete loss of the *Naa20* WT allele expression at mRNA and protein levels (Appendix A).

It has been widely demonstrated that there is a direct correlation between NatB activity, actin cytoskeleton structure and function, and cell proliferation [5,6,7,8,9]. Initially, we validated the activity of NatB in MEFs by demonstrating that inhibition of NatB promotes the disorganization of the actin cytoskeleton (Appendix A), which is consistent with previous reports of NatB activity in various biological systems. As a consequence of this disorganization, there was a decrease in the number of focal adhesions in the central region of the cell and an increase in the number of focal adhesions at the periphery. Furthermore, there was an aberrant expression of several components of the actin cytoskeleton like RhoA and cofilin (Appendix A).

Next, we examined MEFs proliferation rate, and we could observe a marked reduction in *Naa20^−/−^* MEFs proliferation from day 3 post-inactivation, a difference that increased with time (Figure 1A). To determine if cells were arrested at any cell cycle stage, we characterized cell cycle phases using flow cytometry analysis 6 days post *Naa20* inactivation showing that *Naa20^−/−^* MEFs had a lower proportion of cells in S phase, which translated into an accumulation of cells at G1 phase and a slight increase of cells in G2/M phase (Figure 1B). We also examined DNA synthesis by assessing the incorporation of EdU six days after the inoculation of recombinant adenovirus. Our observations revealed a significant decrease in EdU incorporation in *Naa20^−/−^* MEFs compared to the WT (Figure 1C), indicating that Naa20 inactivation leads to a reduction in DNA synthesis.

### 2.2. Hippo Pathway Is Suppressed in Naa20^−/−^ MEFs

The regulation of Hippo–YAP signaling is largely dependent on the actin cytoskeleton, which controls the function of the YAP transcription factor and plays a crucial role in cell proliferation and survival. Recent studies have demonstrated that this pathway is mechanoregulated, with the activation of YAP/WWTR1 requiring actin cytoskeleton activity. Conversely, perturbations to the actin cytoskeleton, whether genetic or pharmacological, have been shown to suppress YAP/WWTR1 activity [33,34]. As Naa20 depletion in MEFs can impact cell cycle progression and actin cytoskeleton structure, we investigated the Hippo–YAP signaling pathway under conditions of NatB inactivation.

In the study, it was found that Naa20 inactivation led to dysregulation of several components of the Hippo pathway. The protein levels of Lats1 and Mst1 kinases and YAP/WWTR1 transcription factors were observed to decrease in the *Naa20^−/−^* MEFs (Figure 2A). Additionally, the YAP protein levels and phosphorylated YAP isoform both displayed a decrease. Interestingly, mRNA expression analysis of the deregulated proteins showed an upregulation of *mLats1*, *mMst1*, *mYap*, and *mWwtr1* expression in the *Naa20^−/−^* MEFs compared to the WT cells (Figure 2B). These results suggest that the stability of LATS1, MST1, YAP, and WWTR1 proteins was reduced. YAP and WWTR1 need to be present in the nucleus of the cell to be active. To assess YAP subcellular localization, immunofluorescence images were taken, which showed that nuclear YAP protein levels were reduced compared to wild type MEFs when Naa20 was depleted (Figure 2C). Additionally, transcriptional analysis of three YAP target genes, *mAnkrd1, mCtgf,* and *mIgfbp3*, showed a significant reduction in their expression in Naa20^−/−^ MEFs (Figure 2D).

Our findings suggest that the lack of NAA20 triggers the breakdown of key constituents of the Hippo pathway, leading to a decline in its activity and a decrease in the expression of various YAP target genes. Consequently, the diminished cell proliferation observed in *Naa20^−/−^* MEFs could be attributed to the inhibition of YAP activity.

### 2.3. Depletion of NAA20 Hampers G1/S Transition and Activates the N-Degron Pathway to Regulate the Stability of p16

Subsequently, we opted to examine the primary constituents of the G1/S checkpoint to gain a more comprehensive understanding of how Naa20 depletion results in a decrease in the proportion of cells in S phase and an augmentation in the number of cells in the G1 phase of the cell cycle (Figure 3).

Surprisingly, we found that the expression levels of most G1/S checkpoint components supported a normal transition from the G1 to the S phase, as explained below (Figure 3A). There was a notable reduction in p53 protein levels in NAA20 deficient MEFs despite the overexpression of *mTp53* mRNA (Figure 3B). Additionally, the mRNA expression of several p53 target genes such as *p21, CCND1, Mdm2, Bbc3*, and *Gadd45α* was upregulated (Figure 3C,D). As a result, p21 and CCND1 protein levels increased upon depletion of NAA20. Interestingly, the upregulation of CCND1 was associated with an increase in CDK6 protein and mRNA expression. Furthermore, in *Naa20^−/−^* MEFs, p16 protein levels decreased, but its transcription was activated. CyclinD1/Cdk6 and CyclinE/Cdk2 complexes promote cell cycle progression from G1 to S phase, and their activity can be inhibited by p21, which was upregulated in the absence of NAA20.

CyclinD1/Cdk6 and CyclinE/Cdk2 complexes promote cell cycle progression from G1 to S phase, and their activity can be inhibited by p21, which was upregulated when NAA20 was depleted. Thus, the phosphorylation status of retinoblastoma protein (pRB) was analyzed to determine if the checkpoint was blocked. It was discovered that pRb was hyperphosphorylated in the *Naa20^−/−^* MEFs, resulting in E2F1 activation. Several E2F1 target genes, such as *Cdkn2A, Tp53, Cdkn1A, Ccdn1, Cdk6, Mdm2*, and *E2F1*, were upregulated as a result. These findings suggest that E2F1 inactivation is not associated with the blockage of transition from G1 to the S phase.

In recent reports, it has been found that there is a connection between protein N-terminal acetylation and protein stability [16,17,18]. Additionally, there exists a particular branch of the N-degron pathway that controls the stability of certain proteins based on the N-terminal acetylation of their initial residue. In yeast, this modification acts as a destabilizing signal, whereas in humans and plants, it stabilizes proteins [10,11,12,13,14,15]. Interestingly, we have observed that some potential NatB substrates, which are responsible for regulating cell cycle progression, such as p16, p53, and YAP, have reduced protein levels in the *Naa20^−/−^* MEFs despite being transcriptionally upregulated. Hence, to assess if the depletion of NAA20 affects the stability of the three proteins, we suppressed the proteasome activity. The results depicted in Figure 4A illustrate that after 6 h of MG132 treatment to inhibit proteasomes, the levels of p16 and p53 were similar in both WT and *Naa20^−/−^* MEFs, indicating that the decrease in their expression is due to reduced stability. In contrast, YAP stability did not show any significant variation. These findings validate that the deficiency of NatB impacts the stability of specific NatB substrates.

To confirm the impact of NatB deficiency on protein stability via the N-degron pathway, we decreased the expression of the primary ubiquitin E3 ligases (N-recognins)—March6, Ubr4, Cnot4, Ubr1, and Ubr2 using siRNAs. These ligases are known to recognize substrates that are targeted for degradation by the N-degron pathway (Figure 4B and Appendix A). It was observed that downregulation of Ubr4 led to an increase in the amount of p16 protein in MEFs compared to scramble-treated cells. Furthermore, when Ubr4 expression was silenced, p16 protein levels were normalized in *Naa20^−/−^* MEFs, showing similar levels to those observed in WT MEFs that were similarly treated. Interestingly, YAP protein levels were also recovered when Ubr4 N-recognin expression was inhibited, although it was not affected when the proteasome was inhibited. This suggests that YAP downregulation is associated with the action of the N-degron pathway. In contrast, p53 protein levels were not recovered despite being sensitive to proteasome inhibition, indicating the involvement of a different ubiquitin E3 ligase in its degradation.

Following the observation that Ubr4 inhibition restored p16 and YAP protein levels in the *Naa20^−/−^* MEFS, we investigated whether silencing Ubr4 could also restore cell proliferation in *Naa20^−/−^* MEFs. According to the data presented in Appendix A, inhibiting Ubr4 reduced cell proliferation in WT MEFs, but it did not have an effect on *Naa20^−/−^* cell growth when compared to control *Naa20^−/−^* MEFs. Further analysis through Western blot confirmed that Ubr4 inhibition recovered p16 and YAP levels, but not p53 and p21, nor pRb phosphorylation levels. Therefore, even with the restoration of p16 and YAP levels, it was not sufficient to normalize *Naa20^−/−^* MEFs proliferation, indicating that other proteins such as p53 that still are deregulated after Ubr4 downregulation, are limiting cellular proliferation.

### 2.4. Depletion of NAA20 Limits DNA Replication Initiation Generating dsDNA Breaks

The FACS analysis revealed that *Naa20^−/−^* MEFs exhibited a reduction in EdU incorporation, indicating a blockade in DNA synthesis. To further investigate the effect of NAA20 depletion on DNA replication, fiber assay analysis was used. This analysis measures the incorporation of thymidine analogues CIdU and IdU to assess DNA synthesis progression (Figure 5A). The results were consistent with the FACS analysis, as there was a lower number of DNA fibers that incorporated either or both molecules when Naa20 was inactivated (Figure 5B). Interestingly, *Naa20^−/−^* MEFs showed a higher total fiber length and fork rate compared to WT MEFs (Figure 5C,D), suggesting a higher processivity when NAA20 is absent. These findings suggest that the defect in DNA replication initiation is the limiting factor that inhibits DNA synthesis in *Naa20^−/−^* MEFs.

Studies have linked a decrease in the number of replication forks and an increase in the replication fork speed with replication stress and DNA damage response (DDR) [35,36]. In this study, we investigated the activation of DNA damage signaling by analyzing H2A.X-S139 phosphorylation (γ-H2A.X) six days after *Naa20* inactivation. Our findings showed that *Naa20^−/−^* MEFs had increased levels of γ-H2A.X compared to WT MEFs as demonstrated by Western blot analysis of total and γ-H2A.X (Figure 6A). Immunofluorescence analysis of MEFs also confirmed the accumulation of γ-H2A.X foci in the nuclei of NAA20-depleted MEFs (Figure 6B). In Naa20^−/−^ MEFs, the DDR is found to be active, which can be linked to the existence of double-strand breaks (DSB) [37]. To evaluate DNA integrity at the single-cell level, we conducted a neutral comet assay. The results shown in Figure 6C confirmed the presence of dsDNA breaks in *Naa20^−/−^* MEFs, as evidenced by a higher tail moment than WT cells.

Based on the data collected, it can be inferred that the disruption of Naa20 activity leads to an imbalance in DNA synthesis dynamics, which is accompanied by dsDNA breaks, ultimately resulting in the activation of the DDR.

### 2.5. Naa20^−/−^ MEFs Growth Arrest Drives to Cellular Senescence

The process of cellular senescence involves a series of steps that result in an irreversible halt in the cell cycle due to stress or damage caused by external factors [31]. Our research has shown that the deactivation of NAA20 is linked to a decrease in cell proliferation, which is associated with replication stress, DSB, and DDR activation. Senescent cells exhibit distinct morphological and molecular features that set them apart from other non-dividing cell types. These cells are flat, enlarged, and may have multiple or enlarged nuclei, indicating a change in shape and granularity. To evaluate cell size and intracellular complexity, we initially used FACS to analyze WT and *Naa20^−/−^* MEFs by measuring the light scattered by the cells. In our study, we noticed that MEFs lacking NAA20 exhibited changes in cell morphology, including an increase in size and complexity, as well as an increase in nuclei (Figure 7A). These changes suggest that NAA20 inactivation affects cell morphology. One of the hallmarks of senescent cells is increased lysosomal activity, which leads to an increase in senescence-associated β-Galactosidase (SA-β-Gal) enzymatic activity, a widely used marker for cellular senescence. Our results showed that six days after NAA20 inactivation, MEFs had higher SA-β-Gal activity compared to WT cells, as depicted in Figure 7B, supporting the idea that cellular senescence is induced in these cells.

In senescent cells, various nuclear structural changes occur, including the presence of DNA-SCARs (DNA segments with chromatin alterations reinforcing senescence) such as the γ-H2A.X foci mentioned earlier (Figure 6B). Additionally, a common nuclear alteration associated with senescence is the decrease in expression of the protein Lamin B1, which is a key player in maintaining nuclear integrity. However, in this study, there was no reduction in mLmnb1 gene expression detected (Figure 7C); on the contrary, an increase in LaminB1 protein levels was observed (Appendix A). The study found that the location of LaminB1 on the nuclear lamina structure was not different in *Naa20^−/−^* MEFs, as observed through immunofluorescence (Figure 7C). However, fragments of nuclear DNA were detected in the cytoplasm close to the nucleus of *Naa20^−/−^* MEFs, similar to what has been found in senescent cells. The nuclear envelope structure of LaminB1 appeared normal while the number of cells with cytoplasmic chromatin was quantified, revealing that almost 15% of *Naa20^−/−^* MEFs had cytosolic chromatin, while less than 2% of WT MEFs had the same (Figure 7D).

The secretion of Senescence-Associated Secretory Phenotype (SASP) factors by senescent cells is a phenomenon observed in certain cases. The SASP factors consist of chemokines, growth factors, metalloproteases, and extracellular vesicles. Some of these factors, such as Interleukin 1-α (mIl1a), Interleukin 6 (mIl6), and chemokine (C-X-C motif) ligand 1 (mCxcl1), are expressed at higher levels in senescent cells. The mRNA expression of *mIl1a* and *mCxck1* was observed to increase in *Naa20^−/−^* MEFs, indicating the activation of SASP factors, while no differences were observed in *mIl6* expression (Figure 7E).

According to the evidence (Figure 7F), mTOR activity is necessary for coordinating cellular senescence, and autophagy is commonly increased in senescent cells. Therefore, we examined various mTOR targets and autophagy components in both WT and *Naa20^−/−^* MEFs. It is noteworthy that mTOR activity is significantly reduced (as evidenced by the decrease in phosphorylation of p70S6K, S6RP, and Ulk1) upon NAA20 depletion. Furthermore, the LC3B-II/LC3B-I ratio is higher in *Naa20^−/−^* MEFs, indicating that autophagy is activated in these cells. This is further supported by the decreased phosphorylation of Ulk1 Serine 757. In cells where NAA20 is depleted, the observed growth arrest is correlated with a decrease in p70S6K and S6RP phosphorylation. This decrease in phosphorylation is also linked to the senescent phenotype observed in *Naa20^−/−^* MEFs, which exhibit various features associated with this phenotype, including increased cellular and nuclear size, SA-β-Gal activity, cytoplasmic chromatin, SASP phenotype, and autophagy induction.

Based on the generated data, it can be inferred that the depletion of NAA20 results in growth arrest due to impaired DNA replication initiation, which leads to double-strand breaks (DSB) and subsequent activation of senescence in MEFs.

## 3. Discussion

Our study demonstrates that the depletion of NAA20 activates the senescence program in MEFs by disrupting the essential role of NatB catalytic subunit in DNA replication initiation and maintaining DNA integrity. Protein N-terminal acetylation is a commonly occurring post-translational modification that plays a vital role in various biological functions. Previous research has linked NatB activity to cellular proliferation, exhibiting delayed or blocked cell cycle progression in yeast, plants, and tumor cells when the enzyme is downregulated or inactivated [5,9,38]. However, the underlying reasons for this phenotype can vary among different organisms.

Protein N-terminal acetylation plays a crucial role in actin cytoskeleton function, as evidenced by its disruption in yeast, Drosophila, and tumor cells upon NatB subunits downregulation or inactivation [5,6,7,8]. Notably, this phenotype is primarily attributed to reduced tropomyosin activity resulting from NatB-mediated N-terminal acetylation deficiency [5,6,7,8]. Our study demonstrates that NatB’s catalytic subunit is also essential for maintaining the actin cytoskeleton structure in MEFs, highlighting the enzyme’s universal significance in actin cytoskeleton function. Moreover, the actin cytoskeleton regulates cell proliferation at various transition points, including G1/S and mitosis [39,40]. Additionally, the F-actin cytoskeleton’s conformation and function are responsible for the activation of various genes that promote DNA replication, mitosis, and checkpoint progression [41]. Moreover, the F-actin cytoskeleton’s regulation of YAP/WWTR1 activity has been established, which indirectly affects the transcription of other master transcription factors. Furthermore, YAP activity requires receptors and structures present on the plasma membrane, such as focal adhesions, to avoid its degradation [42]. Therefore, we can link the downregulation of YAP observed when NatB is inactivated to the reduction in focal adhesions and reorganization of actin stress fibers in Naa20^−/−^ MEFs. Accordingly, it is observed that there is a decrease in YAP and WWTR1 protein levels, along with a reduction in the expression of YAP target genes. Intriguingly, this reduction is linked to the downregulation of various components of the Hippo pathway, including Mst1 and Lats1 despite an upregulation in the expression of Yap, Wwtr1, Mst1, and Lats1 mRNAs by unknown molecular mechanisms. The downregulation of Lats1 is consistent with the accumulation of cells in the G1 phase in NAA20-depleted MEFs, as Lats1 is degraded by the APC/C^Cdh1^ during the G1 phase to facilitate the transition from G1 to S phase [43]. In addition, several studies have demonstrated that the inactivation of NatA and NatB leads to decreased stability of their substrates. These proteins are recognized by different N-recognins, which facilitate their degradation by the proteasome [14,15,44,45,46,47]. Interestingly, YAP, WWTR1, and MST1 are NatB substrates based on their second amino acid, and our findings reveal that downregulation of UBR4 E3 ubiquitin-protein ligase in NAA20-depleted MEFs stabilizes YAP. Hence, our results suggest that inhibition of NatB activity makes YAP, WWTR1, and MST1 susceptible to the N-degron pathway, thereby restraining YAP activity and limiting cell cycle progression. Furthermore, our study indicates that NAA20 inactivation destabilizes p16 and p53, two key components of the G1/S checkpoint and potential NatB substrates, as their stabilization is observed when proteasome activity is inhibited in *Naa20^−/−^* MEFs. Furthermore, we found that the N-degron pathway is responsible for promoting the degradation of p16 when NAA20 is depleted. We observed that UBR4 E3 ubiquitin-protein ligase downregulation increases p16 stability in *Naa20^−/−^* MEFs. However, we did not find any N-recognins that recognize p53. Instead, we found that Mdm2 E3 ubiquitin-protein ligase, which regulates cell cycle progression by promoting p53 degradation [48], is upregulated in the absence of NatB catalytic subunit, which suggests that p53 could be destabilized by Mdm2 when NAA20 is depleted. Nt-acetylation was initially discovered as a protein modification that targeted proteins for proteolysis [23], but recent reports suggest that it can promote protein stability through both N-degron and non-N-degron pathways [14,15,44,45,46,47]. The number of E3 ubiquitin-protein ligases involved in recognizing non-acetylated N-termini is also expanding, including E3 ligases not previously identified as N-degron N-recognins, such as the IAPs [44]. In Drosophila, one NatB substrate was found to be destabilized by the UBR4 E3 ubiquitin-protein ligase when NatB activity was reduced [11], similar to what was observed with p16 and YAP. This suggests that UBR4 N-recognin may play a role in regulating the stability of NatB substrates. There are multiple E3 ubiquitin-protein ligases involved in controlling the stability of cell cycle regulators when NAA20 is depleted in MEFs. Therefore, it is not surprising that downregulating Ubr4 does not restore cellular proliferation in *Naa20^−/−^* MEFs, even when p16 and YAP levels are normalized.

On the other hand, our findings suggest that disabling Naa20 results in cell cycle arrest linked to changes in DNA replication. MEFs that lack NAA20 display impaired initiation of DNA replication, as demonstrated by reduced nucleotide incorporation observed in FACS and fiber formation assays. However, they exhibit increased DNA polymerase processivity during DNA replication, indicating that DNA elongation remains unaffected. While there were no restrictions at the G1/S checkpoint, despite high levels of p21, the evaluation of checkpoint components revealed phosphorylated pRb, activation of cyclin D1 and E2F1 target genes, and downregulation of p53 in these cells, all of which ensure that cells pass through the G1/S checkpoint and enter the S-phase. In NAA20-depleted MEFs, reduced replication origin initiation, and accumulation of DNA double-strand breaks (DSB) indicate replication stress, as observed through comet assay and histone H2A.X activation and subcellular localization at nuclear foci. Deregulation of DNA synthesis components, factors blocking replication fork, or altered frequency of replication initiation can cause inefficient DNA replication and result in stalled or slow-moving replication forks, rendering them vulnerable to DNA damage and replication stress [49]. In the process of DNA replication, cells activate the replication stress checkpoint to monitor the DNA replication fork’s integrity and prevent DNA damage [50]. This results in the arrest of cell cycle progression, impeding new DNA synthesis. Post-translational modifications, including phosphorylation, sumoylation, and ubiquitination, play a critical role in this response, as evidenced by the degradation of several cell cycle regulators following NAA20 depletion. However, *Naa20^−/−^* MEFs are unable to resolve replication stress, leading to the activation of the senescence program.

Senescence can be triggered by various types of stress, both intrinsic and extrinsic, including mitogenic signals, oncogenic activation, perturbed proteostasis, changes in telomeric sequence and structure, genotoxic stress, and chromatin disorganization [31,51]. When NatB is inactivated, replication stress is triggered, which originates from DNA replication initiation blockade and DNA damage. This replication stress causes cell cycle arrest, which eventually leads to senescence induction. The activation of the p53/p21WAF1/CIP1 and/or the p16INK4A/pRB tumor suppressor pathways mediates cell cycle arrest associated with senescence [51]. In NAA20-depleted cells, only p21WAF1/CIP1 is activated despite p53 absence. However, the prolonged expression of any of the four components is sufficient to induce senescence. It is noteworthy that p21WAF1/CIP1 induction plays a crucial role in initiating senescence-mediated growth arrest in response to various stimuli, as well as in response to DNA damage [52]. Senescence is a complex and dynamic process that involves multiple alterations, resulting in an irreversible proliferation arrest, while maintaining cell viability with significant changes in metabolic activity and gene expression. In this study, it was found that depleting NAA20 in MEFs resulted in various morphological and structural changes that are typically observed in senescent cells. These changes include an enlarged and flattened morphology with nuclear enlargement, as well as the development of senescence-associated secretory phenotype and SA-β-galactosidase activity. Senescent cells are also known to upregulate autophagy, including macroautophagy and chaperone-mediated autophagy, without a decrease in mTORC1 activity [53]. It is worth noting that the depletion of NAA20 in MEFs was observed to trigger autophagy and suppress mTORC1 activity. This finding contradicts other studies that suggest mTORC1 activation leads to senescence or that mTORC1 is present in senescent cells. However, our results are consistent with previous research that has identified NAA20 as one of the acetyltransferases whose depletion induces autophagy and inhibits mTORC1 [54,55]. The functional status of mTORC1 in senescence and its connection with autophagy may vary depending on the senescence type and the cell type being examined. Our study has revealed that the inhibition of NAA20 triggers autophagy in MEFs, which is linked to the inhibition of mTORC1. Although our findings differ from other studies that have reported mTORC1 activation inducing senescence or its presence in senescent cells, they are consistent with previous observations where NAA20 was identified among other acetyltransferases whose depletion activates autophagy and inhibits mTORC1 [56]. However, it is not clear whether the functional status of mTORC1 in senescence and its relationship with autophagy varies depending on the type of senescence and the cell type being studied.

According to our findings, the NatB catalytic subunit, NAA20, plays a crucial role in MEFs by preventing growth arrest and maintaining the cells’ senescent state. In addition, the inactivation of NAA20 hinders DNA replication initiation and leads to DNA damage, which triggers autophagy and prevents genetic rearrangements from being transmitted to daughter cells, ultimately avoiding cellular death. Interestingly, we also discovered that there is post-translational regulation of this process that affects protein stability. Specifically, we observed that several NatB substrates become less stable when NatB is inhibited.

## 4. Materials and Methods

### 4.1. Establishment of Mouse Embryonic Fibroblast Cell Strains

To establish immortalized mouse embryonic fibroblasts (MEFs), we used 12.5–13.5 days post coitum mouse embryos from B6.129P2-Naa20tm1a(KOMP)Wtsi. These mice carry a loxP site in intron 4 and intron 5, which results in the removal of exon 5 upon Cre recombinase activity, leading to the generation of a truncated Naa20 protein lacking the catalytic site. After removing the head and internal organs, mouse embryos were washed with sterile PBS (GIBCO), and incubated with 0.25% trypsin–EDTA (GIBCO) for 30 min at 37 °C. The embryos were then mechanically disaggregated using an 18 G diameter syringe and cultured in a 100 mm cell plate with DMEM medium (Dulbbeco’s Modified Eagle Medium, GIBCO) supplemented with 10% decomplemented fetal bovine serum (FBS) and 1% P/S (GIBCO).

After 2–3 passages, the cells were immortalized by retroviral infection with a recombinant virus expressing the large T antigen of the SV40 virus and zeocin resistance gene to select infected cells. Cells were re-infected 24 h later to enhance the transduction efficacy. Zeocin, the selective antibiotic, was added at 200 μg/mL (INVITROGEN) 3–4 days after infection. The medium was replaced every two days to remove dead cells and refresh antibiotic. From the 100 mm cell plate, clones were selected and amplified. Once established, the clones were used for experimental assays.

### 4.2. Inactivation of Naa20 Gene in Mefs

To inactivate *Naa20* in MEFs, cells were infected with an Adenovirus 5 expressing recombinase Cre under the constitutive promoter CMV (AdCre, MOI 2000) to promote *Naa20* gene inactivation (*Naa20^−/−^* MEFs). MEFs were also infected with an empty Adenovirus 5 (AdEmpty, MOI 2000) to use as a wild-type control (WT MEFs). Cells were plated in 6-well plates (62,500 cells/mL). The day after, cells were infected with AdEmpty or AdCre in DMEM supplemented with 2% FBS and 1% P/S. After 24 h, regular DMEM was added to the cells.

Two days post-infection, cells were trypsinized and replated in 62,500 cells/mL (AdEmpty) and 125,000 cells/mL (AdCre). Five days post-infection, the same procedure was performed with the following cell concentrations—175,000 cells/mL (AdEmpty) and 250,000 cells/mL (AdCre). The next day, cells were harvested for the different experimental assays.

### 4.3. Cell Proliferation Assay

Cells were seeded in 6-well plates and infected the day after with AdEmpty or AdCre, as previously described (Inactivation of *Naa20* gene in MEFs). Then, cells were trypsinized and replated after 2 and 5 days of infection. Cell counting was performed at day 2, 3, 4, 5, and 6 post-infection using a TC20™ Automated Cell Counter (Bio-Rad, Hercules, CA, USA), with 4 wells per condition/per day. The cell number detection was performed according to the manufacturer’s instructions.

### 4.4. Cell Cycle Analysis with Flow Cytometry

To study the cell cycle status of the MEFs, the Click-iT™ EdU Alexa Fluor™ 647 Flow Cytometry Assay Kit (Thermo Fisher, Waltham, MA, USA) was used according to the manufacturer’s specifications. Briefly, cells were incubated with the modified thymidine analogue EdU (5-ethynyl-2′-deoxyuridine) for 2 h at 37 °C and 5% CO_2_ incubator at a final concentration of 10 μM to visualize DNA synthesis. The EdU incorporation was detected with Azide Alexa fluor 647, according to the Invitrogen protocol. FxCycleTM Violet was used to quantify the DNA content. BD FACSCanto II Flow Cytometer (BD BioSciences, Franklin Lakes, NJ, USA) was used to acquire the data, and FlowJo software (version 7.6) to process the data.

### 4.5. RNA Extraction and Quantitative Real-Time Pcr Assay

Total RNA was extracted with TRIzol™ reagent according to manufacturer’s instructions. The purified RNA was resuspended in H_2_O treated with 0.1% diethyl pyrocarbonate (H_2_O-DEPC). Reverse transcription was performed as previously described (Balasiddaiah et al., 2013) and quantitative Real-time PCRs (qRT-PCRs) were performed using iQ SYBR Green supermix (Bio-Rad) in a CFX96 Real-Time System (BioRad). Specific primers for each gene were used at a concentration of 15 μM each one. Primer sequences were as follows: mNaa20WT, F 5′CCTTCACCTGCGACGACCTGTT, R 5′-GGAATTCAGGGGCGACAGAG; mGapdh, F 5′GATGGTGAAGGTCGGTGTG, R 5′CTTCCACGATGCCAAAGTTG; mAnkrd1, F 5′TTGGAGAAGCAAGAGGACC, R 5′CAACTGGCAGTTTGTTCTCC; mCtgf, F 5′ATGCTCGCCTCCGTCGCAGG, R 5′GGAAGGACTCACCGCTGCGG; mIgfbp3, F 5′GCAGCCTAAGCACCTACCTC, R 5′CTTGGAATCGGTCACTCGGT; mLats1, F 5′CCTCGTCGAGAGCAGATGTC, R 5′TCCATTGCTTGGGTGAGCTT; mMst1, F 5′CCAGCTTAGGAGAATGGGGTG, R 5′CCGCTGTGTGTAACAGGTTC; mWwtr1, F 5′GGCCCTATCATTCACGGGAG, R 5′TTGACGGTCATGGGTGTCTG; mYap, F 5′CCGACTCCTTCTTCAAGCCG, R 5′TGGAGAGGAGTGAGCTCGAA; mCcdn1, F 5′AGAAGGAGATTGTGCCATCC, R 5′TGGTCTGCTTGTTCTCATCC; mCdkn1A, F 5′GCTGTCTTGCACTCTGGTGT, R 5′GAGGGCTAAGGCCGAAGATG; mCdkn2A, F 5′CGAACTCGAGGAGAGCCATC, R 5′TACGTGAACGTTGCCCATCA; mCdk2, F 5′CTCACGGGCATTCCTCTTCC, R 5′CTGCGGGTCACCATTTCAG; mCdk6, F 5′TTGGATAAAGTTCCAGAGCCCG, R 5′AAAGAGGCTTTCTGCGAAACA; mE2f1, F 5′GCAACTGCTTTCGGAGGACT, R 5′GGTCTTCCCAGGGCTAATCC; mMdm2 F 5′TGTGTTTGGAGTCCCGAGTT, R 5′CTGCTCTCACTCAGCGATGT; mTp53, F 5′CACCTAGCATTCAGGCCCTC, R 5′GAGGGAGCTCGAGGCTGATA; mCxcl1, F 5′CTGGGATTCACCTCAAGAACAT, R 5′AGGGTCAAGGCAAGCCTC; mIl1α, F 5′CGCTTGAGTCGGCAAAGAA, R 5′GCAGAACTGTAGTCTTCGT; mIl6, F 5′TGATTGTATGAACAACGATGATG, R 5′GACTCTGGCTTTGTCTTTCTTGT; mLmnb1, F 5′ATGAAGAGGAGATCAATGAGAC, R 5′CATACTCAATCTGACGCCC; mCnot4, F 5′TGTTGTAGGTCTGTCACAGCG, R 5′TGTTGTAGGTCTGTCACAGCG; mMarch6, F 5′TATGGGTCCTCAGTGGTGGT, R 5′AAATGGGTAAATCCGCCGGT; mUbr1, F 5′ATCGTGGTGGGATCAGCAAG, R 5′ACAGAGCACACACGTTGGAT; mUbr2, F 5′GTGACACTGAGGCGTGGAAA, R 5′GCATGCAGTAGTAGGTGTCACT; mUbr4, F 5′CAAAGCCCCCGTGTATCTGT, R 5′GCCGGTTCAAATCCTCCTCA.

The amount of each transcript was quantified by the formula: 2ct(mGapdh)−ct(gene), where Ct is the cycle at which the fluorescence signal of the transcript exceeds the background fluorescence.

### 4.6. Protein Extraction and Western Blot Analysis

Cells were harvested and cell lysates were collected using 2× Laemmli Sample Buffer (Bio-Rad) supplemented with 2% Sodium Dodecyl Sulfate (SDS) (Bio-Rad), 0.5 M Tris-HCl pH 7.4, β-mercaptoethanol, and protease inhibitors (1 mM PMSF, 1 μg/mL Aprotinin, 1 mM Orthovanadate and 1 mM Sodium Pyrophosphate (Roche, Basel, Switzerland)). The soluble protein concentration was normalized using Revert staining with REVERTTM Total Protein Stain (Li-Cor, Lincoln, NE, USA), scanned in the Odyssey^®^ Fc Imaging System (Li-Cor) with the 700 mm channel and analyzed with Image Studio Lite (version 5.1, Li-Cor) software.

Protein samples were separated by SDS polyacrylamide gel electrophoresis (SDS-PAGE) and transferred onto nitrocellulose trans-blot membrane (BioRad). To avoid nonspecific interactions, membranes were blocked in 5% BSA or 5% non-fat milk in TBS-T solution (Tris-HCl 50 mM (pH 7.6), NaCl 200 mM, 1% Tween 20) for 30 min at room temperature with agitation. Membranes were then incubated with the following primary antibodies for 1 h at RT: NAT5 (Naa20) (Proteintech, Rosemont, IL, USA, 15807-1-AP, 1:500); GAPDH (Sigma-Aldrich, G8795, 1:5000); TAZ (Cell Signaling, Danvers, MA, USA, #8418, 1:1000); YAP (Cell Signaling, #14074, 1:1000); phospho-YAP(Ser397) (Cell Signaling, #13619, 1:1000); LATS1 (Cell Signaling, #3477, 1:1000); Mst1 (Cell Signaling, #3682, 1:1000); Sav1 (Cell Signaling, #13301, 1:1000); p53 (Santa Cruz Biotech, Dallas, TX, USA, sc-99, 1:200); p21 (Santa Cruz Biotech, sc-397, 1:200); p16 (Santa Cruz Biotech, sc-1207, 1:200); Cdk6 (Santa Cruz Biotech, sc-177, 1:200); Cyclin D1 (Cell Signaling, #2926, 1:1000); Cdk2 (Cell Signaling, #2546, 1:1000); Phospho-Rb (Ser807/811) (Cell Signaling, #8516, 1:1000); Phospho-Histone H2A.X(Ser139) (Cell Signaling, #9718, 1:1000); Histone H2A.X (Cell Signaling, #7631, 1:1000); and Lamin B1 (Santa Cruz Biotech, sc-374015, 1:200). After 3 steps of washing with TBS-T, a second incubation was performed for 1 h at room temperature with the secondary antibody conjugated with peroxidase: Anti-Rabbit IgG, HRP-linked Antibody (Cell Signaling, #7074, 1:5000); Anti-Mouse IgG, HRP-linked Antibody (Cell Signaling, #7076, 1:5000); and Anti-Goat IgG-Peroxidase (Sigma-Aldrich, A5420, 1:5000). Chemiluminescence detection was performed using the ECL Ultra detection system (Lumigen, Southfield, MI, USA) and an Odyssey^®^ Fc Imaging System (LI-COR). Signal quantification was performed with the Image Studio Lite (version 5.1, Li-Cor).

### 4.7. Fiber Assay

At day 6 post-inactivation, MEFs were pulse-labeled for 20′ with 25 μM thymidine analogue 5′-chloro-2′-deoxyuridin (CIdU, Sigma), washed three times with pre-warmed PBS, and, subsequently incubated with 250 μM 5′-iodo-2′-deoxyuridine (IdU, Sigma) for 20 min. Labelled cells were collected and resuspended in PBS at a density of 2.5 × 105 cells/mL, and DNA fibers were spread as previously described [1]. After air-drying, DNA was fixed with a 3:1 methanol–acetic acid solution for 5 min at room temperature. Then, slides were air-dried and stored at 4 °C until staining.

For immunolabelling, slides were rehydrated with ddH2O, and then fibers were denatured in a 2.5 M HCl solution for 45 min at room temperature. Slides were then rinsed four times with PBS and incubated 1 h at room temperature in blocking buffer (1% *w*/*v* BSA, 0.1% Triton X-100, PBS). After the blocking step, slides were incubated overnight at 4°C with primary antibodies diluted in blocking buffer (1:100 rat monoclonal anti-BrdU BU1/75 (ICR1) (Abcam, Cambridge, UK, ab6326) [for CIdU] and 1:100 mouse monoclonal anti-BrdU clone B44 (BD Bioscience, 347580) [for IdU]). Then, slides were rinsed three times in PBS before the incubation with the secondary antibodies diluted in blocking solution (1:300 Alexa fluor 594 anti-rat (Invitrogen, A21434) [for CIdU] and 1:300 Alexa Fluor 488 anti-mouse (Invitrogen, Waltham, MA, USA, A21202) [for IdU]) for 1 h at room temperature. After that, slides were rinsed three times in PBS and incubated for 30 min at room temperature with mouse monoclonal anti-single stranded DNA (Clone 16–19, Millipore, MAB3032, 1:100) and goat anti-mouse Alexa Fluor 647 (Invitrogen, A28282, 1:300) diluted in blocking solution. Finally, slides were rinsed in PBS, dried, mounted with antifade mounting medium (Prolong), and sealed with a cover glass. For visualization, a 40× objective in the Axio Imager M1 microscope (Zeiss, Jena, Germany) was used, and fibers were measured using ImageJ software (version 1.48).

### 4.8. Comet Assay

Glass microscope slides were coated with 1% of low melting agarose (Condalab) and air-dried on a flat surface. A 1.2% agarose was prepared and placed at 37 °C before use with the cells. At day 6 post-inactivation, cells were trypsinized and resuspended in 1x PBS at a concentration of 2 × 105 cells/mL. Next, 1.2% pre-warmed low melting agarose was mixed with 170 μL of cell suspension at a ratio of 1:1, and the resulting cell-agarose mixture was spread in a thin gel layer on the agarose-coated slides. Slides were then immersed in lysis solution (2.5 M NaCl, 100 mM disodium EDTA, 10 mM Tris base, 200 mM NaOH, pH 10, 1% sodium lauryl sarcosinate, and 1% Triton X-100) at 4 °C overnight. After lysis, slides were incubated for 30 min at 4 °C with neutral electrophoresis solution (100 mM Tris base, 300 mM sodium acetate, pH 9). The slides were then placed in the electrophoretic tank with pre-chilled neutral electrophoresis buffer, and the power supply was set to 1 V/cm and run for 25 min at 4 °C. After electrophoresis, slides were washed with pre-chilled ddH2O for 5 min at room temperature and then immersed in 70% pre-chilled ethanol for 5 min at room temperature. Slides were air-dried overnight and stored at room temperature. Prior to staining, slied were rehydrated with ddH2O for 1 h at 4 °C. Slides were then stained with DAPI at 5 μg/mL for 10 min at room temperature and dried in the dark before covering with a cover glass and ProLong mounting medium. Slides were analyzed using an Axio Imager M1 microscope (Zeiss) 20× and analyzed using automated OpenComet plugin for ImageJ.

### 4.9. Immunofluorescence and Confocal Microscopy

For immunofluorescence experiments, cells were seeded onto 15 mm cover glasses (Menzel-Glaser, Braunschweig, Germany) coated with collagen type I (BD Bioscience) in 6-well plates. After 16–24 h, cells were fixed with 4% paraformaldehyde (PFA, 16% Formaldehyde Solution, Thermo Scientific) for 10 min at room temperature. After rinsing with 1× PBS, cells were permeabilized with 0.1% Triton^®^ X-100 (Sigma) for 10 min at room temperature and then washed with PBS. For protein detection, cover glasses were incubated with the following primary antibodies diluted in 3% BSA TBS-T at +37 °C for 30 min: Lamin A/C (Cell signaling, #4777, 1:100); YAP (Cell signaling, #14074, 1:50); phospho-Histone H2A.X (Ser139) (Cell signaling, #9718, 1:200); and Lamin B1 (Santa Cruz Biotech., sc-374015, 1:50). After 3 washes with PBS, a second incubation was performed for 30 min at +37 °C with the appropriate secondary antibody conjugated with different fluorophores: Anti-Rabbit IgG (H + L) Alexa Fluor™ 488 (Thermo Fisher Scientific, A21206, 1:1500); Anti-Rabbit IgG F(ab’) fragment-CY3 (Sigma Aldrich, C2306, 1:1000); and Anti-Mouse IgG (H + L) Alexa Fluor™ 488 (Thermo Fisher Scientific, A21202, 1:1000). After mounting the coverslip in Vectashield mounting medium with DAPI (Vector Laboratories), samples were stored at 4 °C until visualization. Images were acquired with an Axiovert 200M confocal LSM 510 META Zeiss microscope using a 40× objective.

### 4.10. E3-Ubiquin Ligases Silencing by siRNAs

Cells were seeded in 6-well plates and infected the day after. After 2 days cells were trypsinized and replated as described previously (Inactivation of *Naa20* gene in MEFS). Four days post-infection, cells were trypsinized and replated at a final concentration of 150,000 cells /mL to transfect siRNAs with cells in suspension. The Lipofectamine^®^ RNAiMAX Transfection Reagent (Invitrogen, 13778) was used according to manufacturer’s instructions to transfect the following siRNAs: siRNA Silencer™ Select Negative Control No. 1 (Thermo Scientific, 4390843), Silencer^®^ Select Ubr1 siRNA (Thermo Scientific, 4390771-s75706), Silencer^®^ Select Ubr2 siRNA (Thermo Scientific, 4390771-s104948), Silencer^®^ Select Ubr4 siRNA (Thermo Scientific, 4390771-s87462), Silencer^®^ Select Cnot4 siRNA (Thermo Scientific, 4390771-s79257), and Silencer^®^ Select March6 siRNA (Thermo Scientific, 4390771-s104524). Transfected cells were collected 48 h post-transfection for Western blot and real-time PCR analysis.

### 4.11. SA-β-Galactosidase Detection

For the detection of senescence-associated β-galactosidase activity (SA-β-Gal), cells were fixed with 2% paraformaldehyde (PFA, 16% Formaldehyde Solution, Thermo Scientific) and 0.5% glutaraldehyde for 10 min at room temperature. After rinsing with 1× PBS two times, cells were incubated with the staining solution (Citric acid/Na phosphate buffer 40 mM; K4[Fe(CN)6]·3H2O 5 mM, K3[Fe(CN)6] 5 mM; NaCl 150 mM, MgCl2 2 mM, and XGal 1%) at +37 °C overnight. The next day, cells were rinsed two times with 1× PBS and washed with methanol for 30 s before being air-dried.

### 4.12. Statistical Analysis

The statistical analysis was performed using Prism 9 (GraphPad software). Normality of samples was assessed using the Shapiro–Wilk test. For the statistical analysis of two groups, unpaired Student’s *t*-test was used if the samples presented normal distribution, while Mann–Whitney test was used for non-normal distributions. In the case of the proliferation and cell cycle assays, two-way ANOVA analysis was performed, followed by Tukey’s multiple comparison post hoc test if a significant result was obtained.

## Figures and Tables

**Figure 1 ijms-24-08724-f001:**
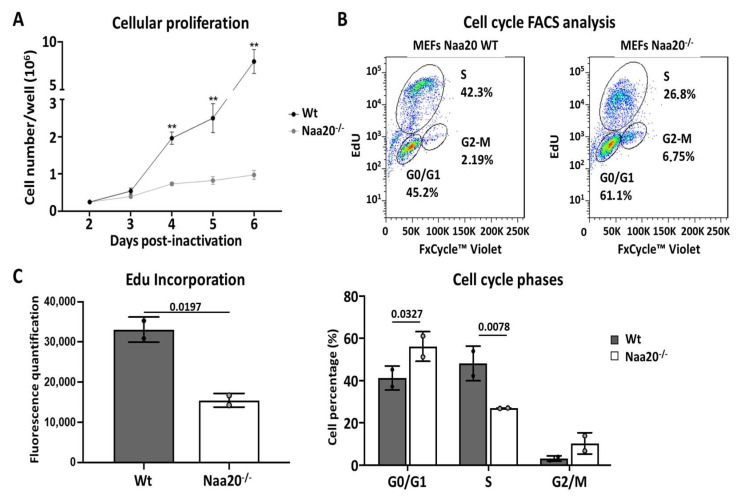
NAA20 inactivation affects cell proliferation and cell cycle progression. (**A**) Proliferation rate analysis in WT and Naa20^−/−^ MEFs counting cells different days post-inactivation. The *p* values were calculated by two-way ANOVA analysis; mean ± SD are represented for each time point with four replicates each. ** *p* < 0.01 (**B**) Flow cytometry analysis 6 days post-inactivation in WT and Naa20^−/−^ MEFs. Quantification of % of cells was performed with FlowJo software (v 7.6), two independent experiments are represented for each condition. The p values were calculated by two-way ANOVA analysis; mean ± SD are represented for each cell cycle phase with two replicates each. (**C**) Quantification of EdU incorporation and cell cycle phases after FACS analysis in WT and Naa20^−/−^ MEFs. Quantification was performed with FlowJo software; two independent experiments are represented for each condition. The *p* values were calculated by unpaired *t*-test; mean ± SD are represented.

**Figure 2 ijms-24-08724-f002:**
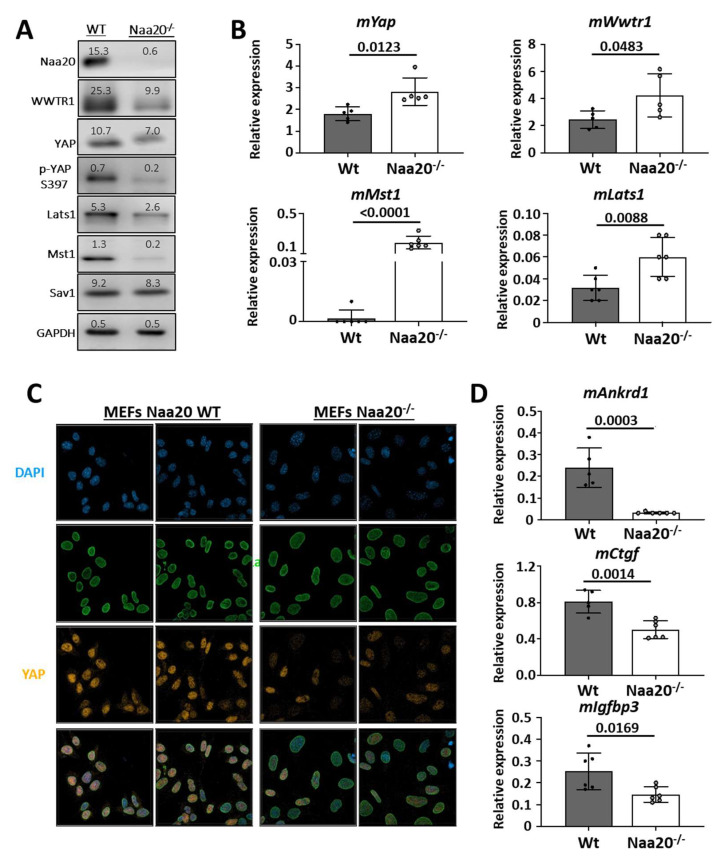
Naa20 inactivation dysregulates Hippo pathway and YAP activity. (**A**) Western blot analysis of WT and *Naa20^−/−^* protein extracts 6 days post-inactivation of Hippo pathway components. The intensity values of each band are indicated. (**B**) Quantitative real-time PCR expression analysis of Hippo pathway components dysregulated at protein level relative to *GAPDH* in WT and *Naa20^−/−^* MEFs 6 days post-inactivation. The *p* values were calculated by unpaired *t*-test; mean ± SD are represented. (**C**) Immunofluorescence (400X) of Lamin A/C (green) and YAP transcription factor (orange) in WT and *Naa20^−/−^* MEFs. Nuclei are marked with DAPI (blue). (**D**) Quantitative real-time PCR expression analysis of Yap target genes relative to *GAPDH* in WT and *Naa20^−/−^* MEFs 6 days post-inactivation. The *p* values were calculated by unpaired *t*-test; mean ± SD are represented.

**Figure 3 ijms-24-08724-f003:**
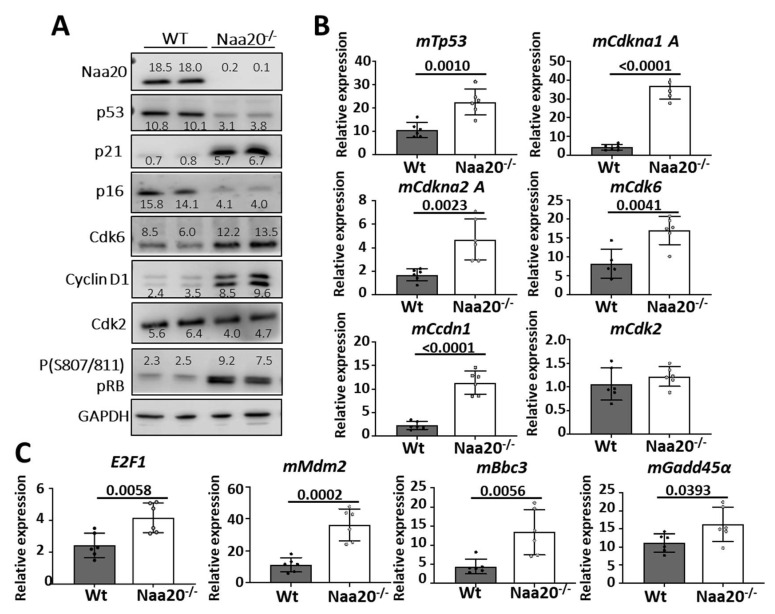
*Naa20* inactivation dysregulates G1 to S phase-transition components but does not affect E2f1 activation. (**A**) Western blot analysis of WT and *Naa20^−/−^* protein extracts 6 days post-inactivation. The intensity values of each band are indicated. (**B**) Quantitative real-time PCR expression analysis of G1 to S transition components relative to *GAPDH* in NAA20 MEFs 6 days post-inactivation. The *p* values were calculated by unpaired *t*-test; mean ± SD are represented. (**C**) Quantitative real-time PCR expression analysis of *mE2F1*, *mMdm2*, *mBbc3* and *mGadd45α* relative to *GAPDH* in NAA20 MEFs 6 days post-inactivation. The p values were calculated by unpaired *t*-test; mean ± SD are represented.

**Figure 4 ijms-24-08724-f004:**
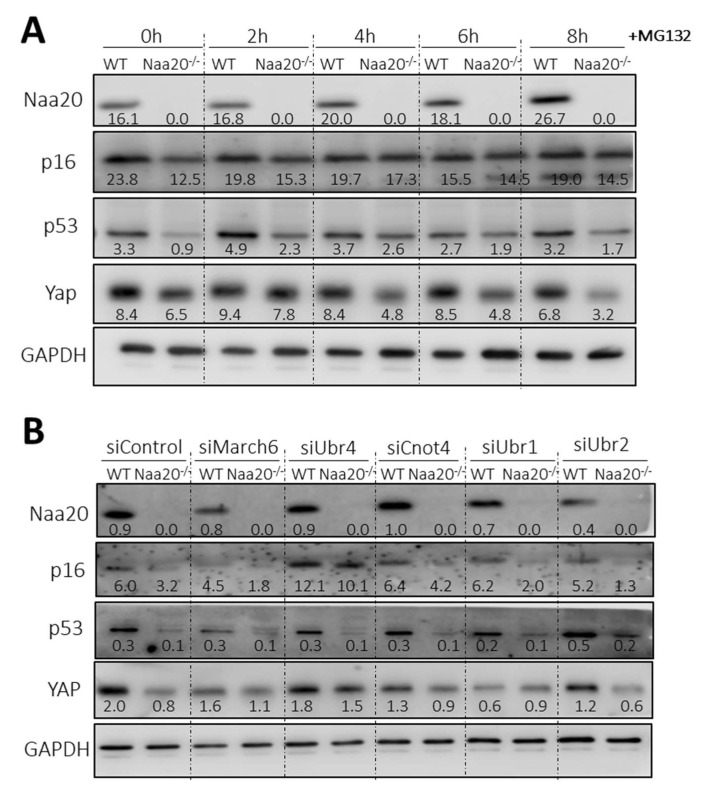
*Naa20* inactivation reduces protein stability of NatB substrates mediated by Ubr4 through the N-degron pathway. (**A**) Proteasome inhibition in NAA20 MEFs with 5 µM MG132 treatment. Western blot analysis of cell extracts of WT and *Naa20^−/−^* cells at basal condition (0 h) and 2, 4, 6, and 8 h after MG132 treatment. (**B**) N-recognins silencing in Naa20 MEFs. March6, Ubr4, Cnot4, Ubr1, and Ubr2 ubiquitin ligases were silenced in WT and *Naa20^−/−^* MEFs 6 days after NAA20 inactivation with siRNAs, and a siRNA (siControl) was used as control. MEFs were transfected with the respective siRNAs 4 days post-inactivation and collected 48 h post-transfection. Western blot analysis of NatB substrate was performed after silencing March6, Ubr4, Cnot4, Ubr1, and Ubr2 ubiquitin ligases in WT and *Naa20^−/−^* MEFs 6 days after *Naa20* inactivation. A specific siRNA (siControl) was used as control. The intensity values of each band are indicated.

**Figure 5 ijms-24-08724-f005:**
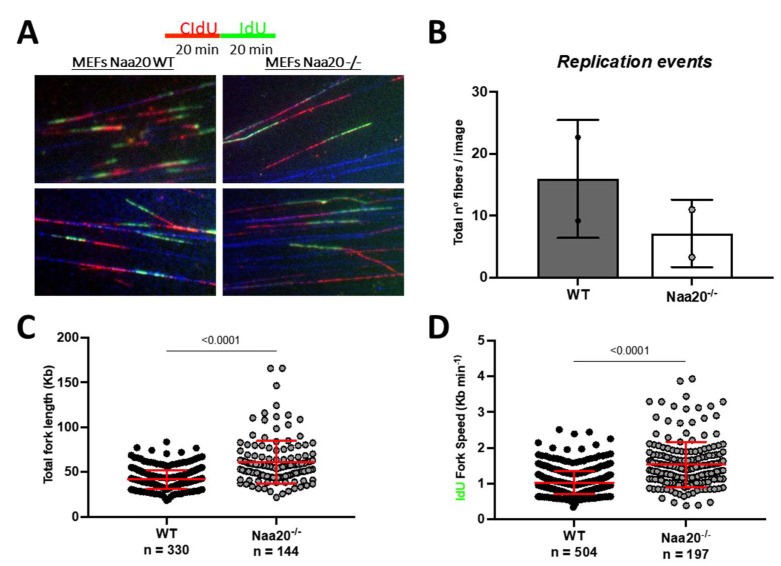
*Naa20* inactivation reduces DNA replication initiation but without affecting replication processivity. (**A**) Representative images of fiber of WT and *Naa20^−/−^* MEFs of DNA fiber assay 6 days post-inactivation. Cells were treated with 25 μM of CIdU for 20 min followed by 250 μM of IdU for another 20 min. (**B**) Number of replication events in two independent experiments represented as number of fibers detected per image. (**C**) Total fork length calculated from the sum of CIdU IdU tracks length. (**D**) Quantification of DNA replication fork rate using IdU track length. The *p* values were calculated by unpaired *t*-test; red lines represent mean ± SD.

**Figure 6 ijms-24-08724-f006:**
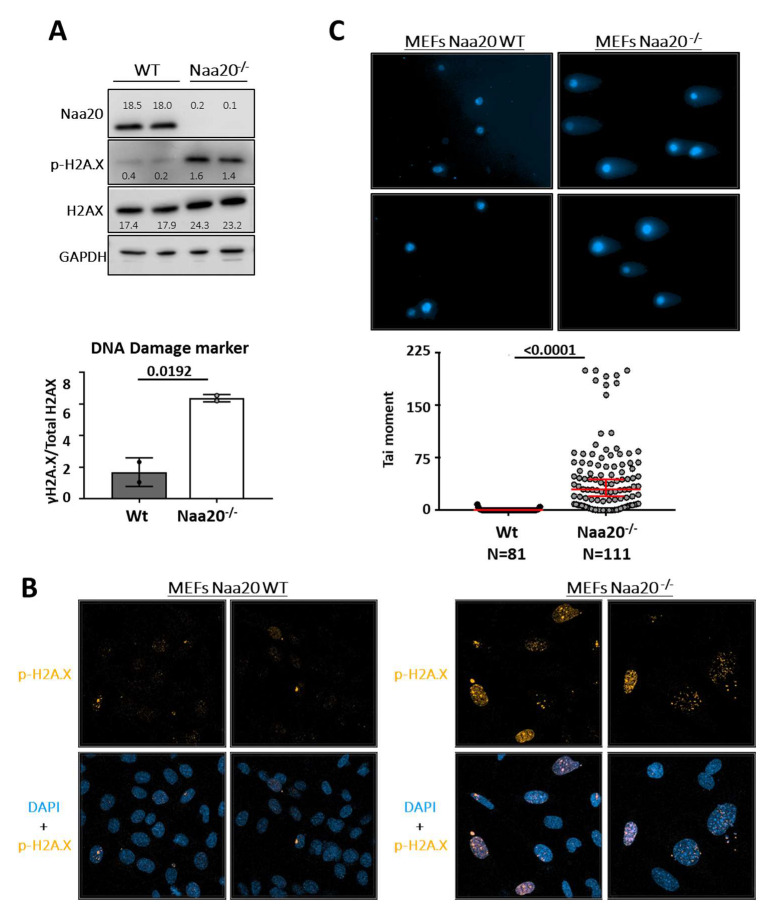
*Naa20* inactivation promotes DNA damage response activation due to double-strand DNA breaks. (**A**) Western blot analysis for the detection of total HA2.X and γ-H2A.X in WT and *Naa20^−/−^* MEFs and quantification of ratio of γ-H2A.X/total H2A.X from immunoblot in arbitrary units. *p* value was calculated by unpaired *t*-test; mean ± SD are represented. The intensity values of each band are indicated. (**B**) Immunofluorescence (400X) of γ-H2A.X (orange) in WT and *Naa20^−/−^* MEFs. Nuclei are marked with DAPI (blue). (**C**) Neutral comet assay in Naa20 MEFs 6 days post-inactivation. With representative images (200X) of neutral comet assay in WT and *Naa20^−/−^* MEFs; and tail moment quantification. The *p* values were calculated by unpaired *t*-test red lines represent mean ± SD.

**Figure 7 ijms-24-08724-f007:**
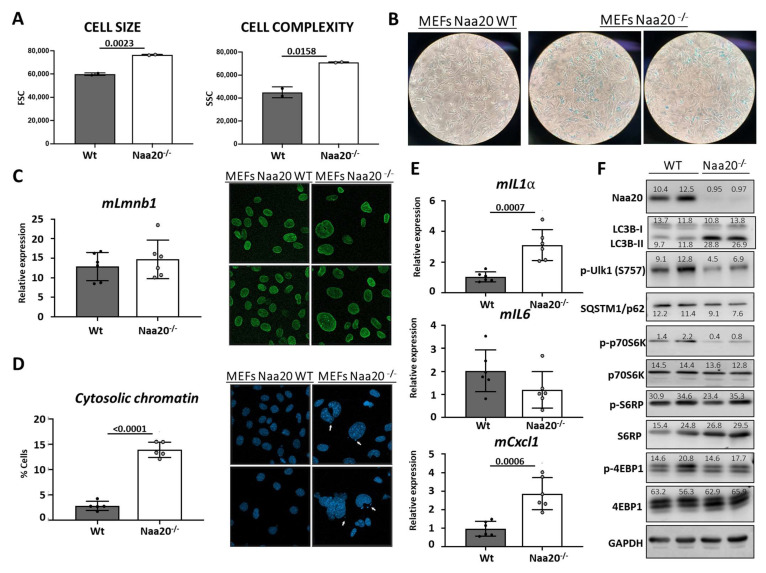
*Naa20* Inactivation is associated with senescence activation in MEFs. (**A**) Cytometric analysis of Naa20 MEFs: cell size (left) and granularity (right) of WT and *Naa20^−/−^* MEFs measured by FSC (forward scatter) and SSC (side scatter), respectively. The p values were calculated by unpaired *t*-test; mean ± SD are represented phase with two replicates. (**B**) β-Gal-SA staining: MEFs were stained to detect β-Gal-SA activity in WT and *Naa20^−/−^* MEFs 6 days after recombinant adenovirus administration (50X). (**C**) Lamin B1 assessment in Naa20 MEFs. Gene expression analysis of Lamin B1 (mLmnb1) by qRT-PCR relative to *GAPDH* (left). The *p* value was calculated by unpaired *t*-test; mean ± SD are represented. Immunofluorescence (400X) of Lamin B1 (green) in WT and *Naa20^−/−^* MEFs (right). (**D**) Cytosolic chromatic assessment in Naa20 MEFs. Nuclei fluorescence staining (400X) with DAPI of WT and *Naa20^−/−^* MEFs (right) and quantification of % cells with DNA (DAPI) blebs in the cytosol in WT and *Naa20^−/−^* MEFs (left). Four independent experiments are represented. The p value was calculated by unpaired *t*-test; mean ± SD are represented of four independent experiments. (**E**) Gene expression analysis of Senescence-Associated Secretory Phenotype (SASP) factors in Naa20 MEFs 6 days post-inactivation. qRT-PCR of mIl1a, Interleukin 1-α. mIl6, Interleukin 6, and mCxcl1, chemokine (C-X-C motif) ligand 1. The *p* values were calculated by unpaired *t*-test; mean ± SD are represented. (**F**) mTOR inactivation and autophagy induction in Naa20 MEFs. Western blot analysis of WT and *Naa20^−/−^* protein extracts 6 days post-inactivation. The intensity values of each band are indicated.

## Data Availability

Not applicable.

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
