# Peer review of "NatB Catalytic Subunit Depletion Disrupts DNA Replication Initiation Leading to Senescence in MEFs"

_ijms, 2023, doi:10.3390/ijms24108724_

Round 1
Reviewer 1 Report
The manuscript by Elurbide et al describes the effect of NatB depletion in mammalian cell cycle progression and suggests a role for the protein in the initiation of DNA replication.
The manuscript reads well and is logically presented. Eukaryotic DNA replication is a fundamental topic in biology so the manuscript could attract many readers. Authors may consider modifying the title a bit to remove the word ‘firing’, that has a very specific meaning in the context of replisome assembly and activation. Furthermore, the acronym MEF may not be clear to everyone.
While the relation between NatB and the cytoskeleton seems to be clear and the authors show data in this direction, it is not so clear what cause the impairment in DNA replication: is it a direct effect because of the lack of acetylation in proteins involved in replication or is it a consequence of cytoskeleton disturbance? This needs clarification in the discussion.
Specific comments:
Page 1 Line 18: instead of firing it would be more appropriate just to write initiation (here and in other places of the text)
Page1 Line 43: remove extra full stop before references (here and in other places of the text).
Page 2 Line 75: correct ‘from both by both’
Page 2 Line 80: remove ‘is’?
Page 3 Line 101: what are chromatin protrusions?
Page 3 Line 118: it can only be intuited that the cell line contains a modified Naa20 gene that can be removed by the Cre recombinase. This needs to be described in the materials or in the results.
Page 3 Line 125: if the correlation between NatB and the cytoskeleton is known, what is the motivation of the authors to perform these experiments. Please explain.
Page 3 Line 132: specify which are the several components.
Page 4 Figure 1: letter size is too small (this applies to all the figures). Panel B is redundant and can be eliminated. Legend to panel A refers to p-values that are not indicated.
Page 4 Line 172: ‘were reduced’ compare to what?
Page 4 Line 173: Figure 2D is referred before than Figure 2C, what is confusing. This can be solved by just swapping panels in the figure itself.
Page 5 Figure 2A: here and in other panels displaying western blots in the manuscript, I suggest to remove the numbers from the photographs and build a bar plot with relative values from the expression of the proteins. If what the authors want is to highlight the variation in expression compared to wt then just normalized to that. As it is now, it is not clear.
Page 6 Line 203: not clear how ‘the analyzed proteins supported a normal transition from the G1 to the S phase’.
Page 6 Line 217: there is no data showing phosphorylation status.
Page 7 Line 231: explain the effect of MG132 in the cells
Page 8 Line 248: ‘decreased the expression’ using siRNA?
Page 11 Line 361: how is the cytosolic chromatin quantified?
Page 12 Line 369: what evidence? If this paragraph is referring to data in figure 7F please indicate in the text.
Page 12 Line 406: The logic of the sentence is not clear: because there is a reduction of focal adhesions YAP is downregulated??
Page 14 Line 511: explain here the particularities of this genotype (Cre recombinase susceptible?)
Page 14 Line 517: include here (P/S) as it is used later
Page 18 Lines710-712: remove or complete
Figure S1: panel B can be split in B and C. Relative expression to what? Explain 99.5% inhibition in the x axis.
Figure S2B: explain how are defined homogenous focal adhesions, include number of cells (not just %), indicate which statistical test is used here.
Figure S3: what is the specific siRNA control
no additional comments
Author Response
Thank you to the reviewer for providing helpful comments that have allowed us to improve the manuscript. We have addressed all of them and have also included additional explanations as requested.
Page 1 Line 18: instead of firing it would be more appropriate just to write initiation (here and in other places of the text)
It has been replaced as suggested
Page1 Line 43: remove extra full stop before references (here and in other places of the text).
Removed
Page 2 Line 75: correct ‘from both by both’
Corrected
Page 2 Line 80: remove ‘is’?
Removed
Page 3 Line 101: what are chromatin protrusions?
We have indicated in the text that these are nuclear blebbs in the cytoplasm detected thanks to the DAPI staining
Page 3 Line 118: it can only be intuited that the cell line contains a modified Naa20 gene that can be removed by the Cre recombinase. This needs to be described in the materials or in the results.
We have described it at the materials and methods section and also in the text.
Page 3 Line 125: if the correlation between NatB and the cytoskeleton is known, what is the motivation of the authors to perform these experiments. Please explain.
We have explained in the text that we wanted to assure that the inactivation of NatB in MEFs reproduced the phenotype observed in different cell types and organisms when this enzyme is inactivated or downregulated as this is the first time this is performed in non-transformed mammalian cells.
Page 3 Line 132: specify which are the several components.
Specified
Page 4 Figure 1: letter size is too small (this applies to all the figures). Panel B is redundant and can be eliminated. Legend to panel A refers to p-values that are not indicated.
We have improved all the figures increasing the letter size. We have eliminated panel B of figure 1 and p values are included
Page 4 Line 172: ‘were reduced’ compare to what?
We have indicated that the reduction is respect wt cells
Page 4 Line 173: Figure 2D is referred before than Figure 2C, what is confusing. This can be solved by just swapping panels in the figure itself.
Panels swapped
Page 5 Figure 2A: here and in other panels displaying western blots in the manuscript, I suggest to remove the numbers from the photographs and build a bar plot with relative values from the expression of the proteins. If what the authors want is to highlight the variation in expression compared to wt then just normalized to that. As it is now, it is not clear.
We have explained in the figure legends the meaning os these values and they are more visible now. We think it is imprtant to visualize at the same time the band and the value to facilitate figure interpretation
Page 6 Line 203: not clear how ‘the analyzed proteins supported a normal transition from the G1 to the S phase’.
We have explained that the levels of expression of the check point proteins analyzed suggest that there is not a G1/S restriction
Page 6 Line 217: there is no data showing phosphorylation status.
We have included the correct information about the antibody as it recognizes the phosphorylated version of pRB
Page 7 Line 231: explain the effect of MG132 in the cells
We have included that it inhibits the proteasome activity
Page 8 Line 248: ‘decreased the expression’ using siRNA?
Added "using siRNAs
Page 11 Line 361: how is the cytosolic chromatin quantified?
We have explained we quantify the cells with nuclear blebs into the cytoplasm
Page 12 Line 369: what evidence? If this paragraph is referring to data in figure 7F please indicate in the text.
Indicated in the text
Page 12 Line 406: The logic of the sentence is not clear: because there is a reduction of focal adhesions YAP is downregulated??
We have explained better that the focal adhesions are requiered to avoid YAP degradation so as there are less focal adhesions this drives to YAP degradation
Page 14 Line 511: explain here the particularities of this genotype (Cre recombinase susceptible?)
We have explained in the the text and material and methods section the presence of two loxP sites flanking exon 5
Page 14 Line 517: include here (P/S) as it is used later
included
Page 18 Lines710-712: remove or complete
removed
Figure S1: panel B can be split in B and C. Relative expression to what? Explain 99.5% inhibition in the x axis.
Splited, we have included that the relative expression is respect GAPDH and that the inhibition is of Naa20 expression when it is inactivated
Figure S2B: explain how are defined homogenous focal adhesions, include number of cells (not just %), indicate which statistical test is used here.
We have explained that regular cells show focal adhesions in the cytosol, nuclear region and cellular periphery (homogenous focal adhesions) and that the inactivation of Naa20 limtis focal adhesions presence to cellular periphery
Figure S3: what is the specific siRNA control
Indicated the reference of the product at the figure legend and at the materials and methods section. It is: siRNA Silencer™ Select Negative Control No. 1 (Thermo Scientific, 4390843).
Reviewer 2 Report
The authors seem to be concluding that the changes seen in the cell cycle/cell proliferation are caused by the cytoskeletal changes. A direct demonstration of this would strengthen the conclusion.
The result of strong activation of p21 levels in the Naa20-/- cells when p53 levels were negligible were surprising. Although there was some discussion about this, an important evaluation would be to see whether KD of p53 in these cells would also still show this increase in p21 levels.
Author Response
Thank you to the reviewer for providing helpful comments.We include below an explanation for the comments presented.
The authors seem to be concluding that the changes seen in the cell cycle/cell proliferation are caused by the cytoskeletal changes. A direct demonstration of this would strengthen the conclusion.
We don't think that the main driver of cell cycle changes are caused by a dysregulation of the actin cytoskeleton. We have shown that YAP downregulation can help to block cell proliferation as it has been presented in many publications and YAP downregulation can be associated to actin cytoskeleton disregulation. We have observed a blockade in DNA replication initiation, which we do not believe can be attributed to actin cytoskeleton disruption. As there are numerous proteins that are NatB substrates, any one of them may be limiting DNA replication initiation when it is not N-terminal acetylated due to reduced stability or activity when the acetyl-group is not linked to the first amino acid. We must investigate further to identify the NatB substrate that is limiting DNA replication when Naa20 is inactivated.
The result of strong activation of p21 levels in the Naa20-/- cells when p53 levels were negligible were surprising. Although there was some discussion about this, an important evaluation would be to see whether KD of p53 in these cells would also still show this increase in p21 levels.
We agree with the reviewer that it is very surprising to observe such upregulation of p21 when p53 is mostly absent. However p21 is a downstream target of several signaling pathways, including p38 MAPK, JNK, and NF-κB (Abbas T, Dutta A. p21 in cancer: intricate networks and multiple activities. Nat Rev Cancer. 2009 Jun;9(6):400-14.). Activation of these pathways can lead to the transcriptional upregulation of p21 and a sustained p21 expression triggers replication stress, DNA damage and senescence (Galanos, P., Vougas, K., Walter, D. et al. Chronic p53-independent p21 expression causes genomic instability by deregulating replication licensing. Nat Cell Biol 18, 777–789 (2016).). There are many examples supporting the biological roles of p53-independent p21 upregulation in apoptosis, tumor growth, senescence and many other biological functions (Fares B, Berger L, Bangiev-Girsh E, Kakun RR, Ghannam-Shahbari D, Tabach Y, Zohar Y, Gottlieb E, Perets R. PAX8 plays an essential antiapoptotic role in uterine serous papillary cancer. Oncogene. 2021 Aug;40(34):5275-5285; Ruan B, Liu W, Chen P, Cui R, Li Y, Ji M, Hou P, Yang Q. NVP-BEZ235 inhibits thyroid cancer growth by p53- dependent/independent p21 upregulation. Int J Biol Sci. 2020 Jan 14;16(4):682-693; Crochemore C, Fernández-Molina C, Montagne B, Salles A, Ricchetti M. CSB promoter downregulation via histone H3 hypoacetylation is an early determinant of replicative senescence. Nat Commun. 2019 Dec 6;10(1):5576.)
Therefore we should explore other signaling pathways that could be driving p21 upregulation when Naa20 is depleted. This is something that we are planning to do.
Reviewer 3 Report
Thank you for sending me this interesting manuscript by Jasmin Elurbide et al. The authors evaluated the Alpha amino terminal acetyltransferase B (NatB), a critical enzyme responsible for acetylating the amino terminal end of proteins. The authors found that depletion of NAA20 results in decreased cell cycle progression and DNA replication firing, ultimately leading to the senescence program. In addition, author found that the inactivation of NAA20 hinders DNA replication firing and leads to DNA damage, triggering autophagy and prevents genetic rearrangements from being transmitted to daughter cells, ultimately avoiding cellular death.
There are issues that are listed in order, as follows:
1) Page 2, line 68 : “The cell cycle is a well-organized, unidirectional process that ensures proper cell di- vision.. [25].” Fix this typo error.
2) page 2, line 75 : “DNA replication can face disruption from both by both endogenous sources, such as reactive oxygen species, and exogenous sources, such as UV” it seems is a duplicate word both, fix this error.
3) page 2, line 93: “research to fully understand the mechanisms underlying cellular senescence. [31,32].” Delete the extra dot before the cites.
4) explain at a better detail why the mLats1, mMst1, mYap and mWwtr1 mRNA expression increase in the Naa20 -/- MEFs , it is interesting that the YAP protein levels have a decrement.
5) in figure 3A, please indicate what the numbers on the gel represent.
6) page 12, 401 : “are responsible for the activation of various genes that promote DNA replication, mitosis, and checkpoint progression [41] Moreover,” add the dot before moreover.
7) please provide figures at higher resolution and change the font size, as some titles are hard to read in some figures as 1, 3, 6, 7.
minor editing are required
Author Response
Thank you to the reviewer for providing helpful comments that have allowed us to clarify some aspects of the manuscript. We have addressed all of them and have also included additional explanations as requested
1) Page 2, line 68 : “The cell cycle is a well-organized, unidirectional process that ensures proper cell di- vision.. [25].” Fix this typo error.
Corrected
2) page 2, line 75 : “DNA replication can face disruption from both by both endogenous sources, such as reactive oxygen species, and exogenous sources, such as UV” it seems is a duplicate word both, fix this error.
Fixed
3) page 2, line 93: “research to fully understand the mechanisms underlying cellular senescence. [31,32].” Delete the extra dot before the cites.
Deleted
4) explain at a better detail why the mLats1, mMst1, mYap and mWwtr1 mRNA expression increase in the Naa20 -/- MEFs , it is interesting that the YAP protein levels have a decrement.
We have mentioned that the upregulation of these four genes is due to unknown mechanisms, as we have not identified in the literature any common pathway responsible for their expression. However, the crucial aspect is that the depletion of Naa20 leads to decreased protein levels in cells, which may be attributed to increased protein instability and not to a downregulation their transcription or mRNA stability.
5) in figure 3A, please indicate what the numbers on the gel represent.
Indicated
6) page 12, 401 : “are responsible for the activation of various genes that promote DNA replication, mitosis, and checkpoint progression [41] Moreover,” add the dot before moreover.
Added
7) please provide figures at higher resolution and change the font size, as some titles are hard to read in some figures as 1, 3, 6, 7.
The figures have been improved